# Financial Toxicity in Japanese Patients with Metastatic Renal Cell Carcinoma: A Cross-Sectional Study

**DOI:** 10.3390/cancers16101904

**Published:** 2024-05-16

**Authors:** Go Kimura, Yasuhisa Fujii, Kazunori Honda, Takahiro Osawa, Yosuke Uchitomi, Miki Kondo, Ariko Otani, Tetsuya Wako, Daisuke Kawai, Yoshihide Mitsuda, Naotaka Sakashita, Nobuo Shinohara

**Affiliations:** 1Department of Urology, Nippon Medical School Hospital, 1-1-5, Sendagi, Bunkyo-ku, Tokyo 113-8603, Japan; gokimura@nms.ac.jp; 2Department of Urology, Tokyo Medical and Dental University, 1-5-45, Yushima, Bunkyo-ku, Tokyo 113-8510, Japan; y-fujii.uro@tmd.ac.jp; 3Department of Clinical Oncology, Aichi Cancer Center, 1-1, Kanokoden, Chikusa-ku, Nagoya 464-8681, Aichi, Japan; khonda@aichi-cc.jp; 4Department of Renal and Genitourinary Surgery, Hokkaido University Graduate School of Medicine, Kita 15 Nishi 7, Kita-ku, Sapporo 060-8638, Hokkaido, Japan; taka0573@med.hokudai.ac.jp; 5Innovation Center for Supportive, Palliative and Psychosocial Care, National Cancer Center Hospital, 5-1-1 Tsukiji, Chuo-ku, Tokyo 104-0045, Japan; yuchitom@ncc.go.jp; 6Department of Nursing, National Cancer Center Hospital East, 6-5-1 Kashiwanoha, Kashiwa 277-8577, Chiba, Japan; mikkondo@east.ncc.go.jp (M.K.); aotani@east.ncc.go.jp (A.O.); 7Department of Pharmacy, Nippon Medical School Hospital, 1-1-5 Sendagi, Bunkyo-ku, Tokyo 113-8603, Japan; t-wako@nms.ac.jp; 8Eisai Co., Ltd., 4-6-10 Koishikawa, Bunkyo-ku, Tokyo 112-8088, Japan; dkawai0128@gmail.com (D.K.); y2-mitsuda@hhc.eisai.co.jp (Y.M.); 9Medilead, Inc., 24F Tokyo Opera City Tower, 3-20-2, Nishishinjyuku, Shinjyuku-ku, Tokyo 163-1424, Japan; n_sakashita@medi-l.com

**Keywords:** renal cell carcinoma, financial toxicity, financial burden, financial distress, health-related quality of life

## Abstract

**Simple Summary:**

We investigated financial toxicity in Japanese patients with metastatic renal cell carcinoma using the COST tool. Despite universal coverage, patients in Japan had similar levels of financial toxicity as in other countries. There was a positive correlation between the FACT-G total score and the COST score. Age < 65 years and not having private health insurance were associated with higher financial toxicity.

**Abstract:**

Information on the financial toxicity experienced by Japanese patients with metastatic renal cell carcinoma (mRCC) is lacking, even though Japan has its own unique public health insurance system. Thus, a web-based survey was conducted to evaluate the financial toxicity experienced by Japanese mRCC patients using the COmprehensive Score for financial Toxicity (COST) tool. This study enrolled Japanese patients who underwent, or were undergoing, systemic therapy for mRCC. The outcomes evaluated were the distribution of COST scores, the correlation between COST and quality of life (QOL) assessed by the Functional Assessment of Cancer Therapy-General (FACT-G) scale, and demographic factors associated with financial toxicity. The median (range) COST score was 19.0 (3.0–36.0). The Pearson correlation coefficient for COST and FACT-G total scores was 0.40. Univariate analysis revealed that not having private health insurance and lower household income per year were significantly associated with lower COST scores. Multivariate analyses showed that age < 65 years and not having private health insurance were significantly associated with lower COST scores. This study revealed that Japanese mRCC patients experience adverse financial impacts even under the universal health insurance coverage system available in Japan, and financial toxicity negatively affects their QOL.

## 1. Introduction

Renal cell carcinoma (RCC) is the 12th most common malignant tumor worldwide, with approximately 431,000 new cases diagnosed in 2020, based on estimates from GLOBOCAN data [1]. In Japan, over 21,000 patients were diagnosed with RCC in 2019 [2]. So far, many anticancer drugs have been approved for the treatment of RCC by the Pharmaceuticals and Medical Devices Agency in Japan, including interferon alfa, tyrosine kinase inhibitors (TKIs), mammalian target of rapamycin inhibitors, and immune-oncology (IO) drugs. In clinical practice, IO/IO and IO/TKI combination therapy and TKI monotherapy are used as first-line treatments [3]. With combination therapy, the direct cost of systemic therapy is higher than with monotherapy.

The importance of evaluating a patient’s financial well-being resulting from direct or indirect treatment costs, including payments for accommodation, food, and transportation, was recently established [4]. This evaluation has given rise to the term financial toxicity, which has been used to describe the personal financial burden faced by patients receiving cancer treatment [5,6]. Previous reports suggested links among financial toxicity, higher mortality, and health-related quality of life (HRQOL) [6,7,8,9]. de Souza et al. [10] developed and validated the COmprehensive Score for financial Toxicity (COST) to measure financial toxicity. For stage IV cancer patients in the US, the study showed that the COST was related to income, psychosocial distress, and HRQOL [10]. For metastatic (m)RCC patients, the factors associated with worse financial toxicity were age < 65 years, high out-of-pocket cost, and receiving and paying for treatment in a real clinical setting (i.e., not participating in a clinical trial) [10]. Another study reported that female sex, younger age, urban living situation, lower educational level, lower income, higher anxiety about contracting COVID-19, having metastatic disease, and a higher distress score about cancer progression were associated with low COST [11].

In Japan, insurance systems differ from those in other countries. All citizens and foreign residents must obtain national health insurance, and up to 30% of all healthcare costs are paid out-of-pocket by the patient. Moreover, citizens who have public health insurance coverage in Japan can apply for a “High-cost Medical Expense Benefit System”, which allows them to reduce the medical costs they pay to a limited monthly amount. Honda et al. [12] surveyed financial toxicity using COST among stage IV Japanese patients who received chemotherapy. They showed that irregular employment, retirement due to cancer, and strategies to cope with the cost of cancer care were associated with worse financial toxicity. In a study of Japanese gynecologic cancer patients who had experienced drug treatment, Kajimoto et al. found a median COST score of 19 [13]. However, for Japanese mRCC patients, data on financial toxicity evaluated using the COST tool are lacking. Additionally, it is unknown if there is an association between financial toxicity and QOL in Japanese mRCC patients. Cella et al. [14] developed a general cancer QOL measure, namely, the Functional Assessment of Cancer Therapy-General (FACT-G). Subsequently, de Souza et al. [10] suggested that, as a QOL measure, the FACT-G correlates more strongly with COST than the European Organization for Research and Treatment of Cancer’s Quality of Life Questionnaire-global health status [15]. Therefore, this study aimed to evaluate financial toxicity for Japanese mRCC patients using the COST tool through a web-based survey and assess QOL using the FACT-G.

## 2. Materials and Methods

### 2.1. Study Design and Participants

This was a cross-sectional web survey study conducted in Japan. The study targeted patients with mRCC who received systemic cancer therapy. Patients who agreed to answer the web questionnaire survey, were ≥20 years old, were living in Japan, and underwent or were undergoing molecular targeted therapy or immune checkpoint inhibitor therapy (including combination therapy) for mRCC were included in this study. Patients and their families engaged in pharmaceutical companies, market research or marketing-related companies, healthcare professionals, and those with multiple primary cancers were excluded.

All data collection by web-based survey was conducted between May 2022 and June 2022. The following survey panels and an online peer support group were used: Research Panel, Inc., Rakuten Insight, Inc., 3H Holdings, Inc., Nikkei Business Publications, Inc., and General Incorporated Association Cancer Parents. 

All methods were carried out per the principles and guidelines of the Declaration of Helsinki. The research ethics committee of the Japanese Association for the Promotion of State-of-the-Art in Medicine approved the study protocol. All the data collected remained confidential and were only used for research purposes. All participants gave informed consent to participate in this study.

### 2.2. Survey Implementation

#### 2.2.1. Participants’ Characteristics

The demographic information of mRCC patients collected included age, gender, treatment history and duration of administration, employment status, insurance type, household income, and household savings.

#### 2.2.2. COST

COST is an 11-item questionnaire, with each item scored using a 5-point Likert scale (0 = not at all to 4 = very much) [10]. Therefore, total COST scoring is distributed on a scale of 0–44; lower COST values indicate worse financial toxicity. For the COST evaluation method, items 2, 3, 4, 5, 8, 9, and 10 in the COST were calculated as reversed scores. The patients answered each item of the COST questionnaire themselves. For this survey, we used the version of COST translated into Japanese and validated by Honda et al. [12].

#### 2.2.3. FACT-G

The FACT-G includes 27 items designed to measure domains of HRQOL in patients with cancer: physical (7 items), social/family (7 items), emotional (6 items), and functional well-being (7 items). Each item is scored using a 5-point Likert scale (0 = not at all to 4 = very much). If half of the items comprising a subscale are answered, a subscale score is computed as the prorated sum of the item responses for that subscale. The FACT-G total score is calculated as the sum of the four subscale scores, provided that the overall item response is at least 80%. The total score of the FACT-G is distributed on a scale of 0–108; higher subscale and total scores indicate better QOL. For the FACT-G evaluation method, the score was calculated according to the FACT-G scoring guidelines [16], and the scores of reverse items were subtracted from 4. If there were missing items, the score was calculated using a proportional distribution method, in which the sum of the scores of the answered items was multiplied by the total number of items in the subdomain and divided by the number of answered items. The patients answered each item of the FACT-G questionnaire themselves. We used Japanese FACT-G ver.4.0 in this study [14].

### 2.3. Outcomes

The outcomes investigated in mRCC patients were the distribution of COST scores, the correlation between COST and QOL assessed by FACT-G, and demographic factors associated with financial toxicity.

### 2.4. Statistical Analysis

The sample size of the target population was set based on the number of patients registered in the web-based survey panels. The target number was 95 patients.

R (version 4.2.0, R Foundation for Statistical Computing, Vienna, Austria) was used to perform all the statistical analysis. Statistical significance was set at a *p* value < 0.05. 

The Pearson correlation coefficient assessed the relationship between the COST score and FACT-G. Univariate analysis, multiple regression analysis, and multivariate analysis were performed. Based on a previous report, we selected seven variables: age (<65 or ≥65 years) [11], type of treatment modality [17], employment status [10,18], treatment duration [19], insurance [17], household income [17], and household savings [12]. We excluded any variables from the multivariate analysis if there was a small number of patients, and the variable was analyzed by univariate analysis.

## 3. Results

### 3.1. Participant Characteristics 

In this study, 508 mRCC patients were sent an e-mail requesting a survey response, irrespective of treatment history. Of these patients, 101 who received systemic treatment responded and provided informed consent to participate in the web survey, resulting in a survey response rate of 19.9% (101/508). Six patients had contradictory answers, and 12 met the exclusion criteria; thus, the data of 83 patients were analyzed in this study (Table 1, Appendix A).

Patients with mRCC had a mean (standard deviation [SD]) age of 44.2 ± 12.2 years, and 63.9% were men. The proportions of patients with a duration of systemic therapy of <6 months were 30.1%; 6 months−2 years, 34.9%; and >2 years, 34.9%. Most of the patients (50 [60.2%]) were in regular employment, with 33 (39.8%) being in non-regular employment. Most patients (48 [57.8%]) did not have private health insurance. The household income per year was not reported by 44.6% of patients, but most of the remaining patients (16.9%) were in the ¥4,000,000−5,999,999 range. Household savings were not reported by 54.2% of patients, but most remaining patients (20.5%) had <¥2 million. Reimbursement system use for high-cost medical care was reported by 59.0% of patients.

### 3.2. Correlation between FACT-G QOL and COST Financial Toxicity

The median (range) COST score was 19.0 (3.0–36.0), and the mean ± SD was 17.8 ± 7.0 (Figure 1). The median (range) FACT-G total score was 61.0 (28.0–89.0). The Pearson correlation coefficient (*r*) for COST and the FACT-G scores was 0.40 (*p* < 0.001). The *r* for COST and physical, social/family, emotional, and functional well-being in the FACT-G subscales were 0.21 (*p* = 0.056), 0.17 (*p* = 0.13), 0.40 (*p* < 0.001), and 0.24 (*p* = 0.030), respectively, indicating a positive correlation (Figure 2).

### 3.3. Linear Regression Analysis of COST

In univariate analyses associated with COST using linear regression (age, treatment history, duration of systemic therapy, employment status, and private health insurance), not having private health insurance (partial regression coefficient (β), −4.23; 95% confidence interval [CI], −7.19 to −1.27; *p* = 0.0057) was significantly associated with a lower COST value, indicating greater financial toxicity. In multivariate analyses of COST using linear regression (age, treatment history, duration of systemic therapy, employment status, and private health insurance), age < 65 years (β, −5.21; 95% CI, −10.4 to −0.0279; *p* = 0.049) and not having private health insurance (β, −4.90; 95% CI, −8.02 to −1.79; *p* = 0.0024) were significantly associated with lower COST values, indicating higher financial toxicity (Table 2). Employment status (β, −2.89; 95% CI, −5.93 to 0.149; *p* = 0.062) tended to affect COST values. Univariate analyses were conducted for household income per year and household savings, which were not included in the multivariate analysis because only a few responses (*n* = 46 and *n* = 38, respectively) were obtained. These showed that lower household income per year was significantly associated with lower COST scores, indicating higher financial toxicity; <¥2,000,000 (β, −11.7; 95% CI, −20.7 to −2.68; *p* = 0.012), ¥2,000,000−3,999,999 (β, −12.3; 95% CI, −18.7 to −5.78; *p* = 0.00045), and ¥4,000,000−5,999,999 (β, −9.52; 95% CI, −15.4 to −3.61; *p* = 0.0023).

## 4. Discussion

To the best of our knowledge, this study is the first to identify factors affecting financial toxicity for Japanese patients with drug-treated mRCC using the COST tool. In a previous study of patients diagnosed with mRCC in Japan between 2008 and 2018, the mean age of patients was reported to be 66 years (range: 37–87 years) [20]. In comparison, patients in the present study were recruited via the Internet and thus included a younger patient population, with a mean age of 44.2 years. In this age group, illness is considered to have a greater impact than in other age groups because of how it affects work and child-rearing, so this study is unique in that it analyzed financial toxicity in this age group.

In this study, the median COST value was 19.0 (range 3.0–36.0), which was consistent with those reported for cancer patients in Japan and mRCC patients overseas. For example, Honda et al. [12] evaluated financial toxicity in Japanese cancer patients and reported that the median COST score was 21 (range 0–41), while Ezeife et al. [11] evaluated financial toxicity in Canadian mRCC patients and found a median COST score of 20.5 (range 3–44). Staehler et al. evaluated financial toxicity in US mRCC patients, and the median COST score was 22 (range 4–36) [21]. This suggests that mRCC patients in Japan experience a similar level of financial toxicity as those in other countries, even though Japan has a universal health insurance system and 59.0% of the patients in this study were covered by the high-cost medical care reimbursement system, which limits payment for cancer treatment to a maximum amount.

de Souza et al. [10] reported that COST was related to psychosocial distress and HRQOL, with a median FACT-G score of 79 (range 23–108), and there was a positive correlation between COST and FACT-G (*r* = 0.42; *p* < 0.001). Pangestu et al. [22] systematically evaluated the association between COST and HRQOL via a meta-analysis, which showed that a reduction in financial toxicity may contribute to improved HRQOL. In our study, there was a moderate positive correlation between the COST score and FACT-G (*r* = 0.40; *p* < 0.001), similar to the findings of de Souza et al. [10]. When the correlation between the subscales of FACT-G and the COST score was evaluated, “emotional well-being” (*r* = 0.40; *p* < 0.001) had a larger correlation coefficient than the other three subscales: “physical well-being” (*r* = 0.21; *p* = 0.056), “social/family well-being” (*r* = 0.17; *p* = 0.13), and “functional well-being” (*r* = 0.24; *p* = 0.030). This is similar to the “psychosocial distress” findings reported by de Souza et al. [10]. It is likely that in this study, the COST questionnaire demonstrated this correlation because of the way the questions focusing on the emotional aspects of financial toxicity were asked.

The factors influencing higher financial toxicity in cancer patients have been reported [11,12,17,19,23]. In this study, factors influencing higher financial toxicity were age < 65 years and not having private health insurance, with the result for age < 65 years similar to that reported by Ezeife et al. [11]. It can be assumed that the younger age group has lower annual income and savings, as well as higher costs of living, such as costs involved in child rearing, which may contribute to the higher financial toxicity. In Japan, 42.6% of the population (men: 43.2%, women: 42.2%) was enrolled in cancer insurance or had a cancer rider within a private insurance policy in 2019, with the rate increasing year by year [24]. The enrollment rate by age group was 25.4% for those in their 20s, 46.4% for those in their 30s, 50.8% for those in their 40s, 44.7% for those in their 50s, and 40.3% for those in their 60s. The enrollment rate tends to be high among those in their prime working years and raising children and low among the younger generation. When enrollment rates by annual household income were examined, 26.9%, 44.1%, 54.7%, 52.9%, and 51.9% of those with annual household incomes of less than ¥3 million, ¥3,000,000–4,999,999, ¥5,000,000–6,999,999, ¥7,000,000–9,999,999, and more than ¥10 million, respectively, were enrolled. Higher annual household incomes tended to be associated with higher rates of cancer insurance coverage, perhaps because of greater financial resources. Possible reasons for the significant influence of having private health insurance in the present study may be that many of the eligible patients had annual household incomes of <¥6 million and savings of <¥2 million. It has been previously reported that not having insurance coverage for cancer-related costs was associated with financial toxicity [17]. Conversely, “non-regular employment”, which was a factor reported by Honda et al. [12] did not significantly differ in this study but showed a trend toward higher financial toxicity. Additionally, in the systematic review and the domestic report by Nezu et al. [19], the duration of treatment (cycles of drug therapy) was mentioned as a factor affecting financial toxicity, but in the present study, no significant difference was found. “Lower household income” is an important factor for higher financial toxicity [25,26,27] and “greater household savings” is an important factor for lower financial toxicity [12]. In this study, a significant difference in household income was observed for the group below ¥6 million, while no significant difference was observed for household savings. The reason for this may be that the mean age of the patients in this study was 44.2 years, and their household savings were small, so household income may have had a greater influence on financial toxicity than household savings.

Various efforts are being considered as solutions to the financial burden, including the expansion of social safety nets (i.e., medical insurance, pension, unemployment insurance, job seeker support programs, welfare payments, among others), the reform of drug pricing regulations, expanding access to insurance, and screening for financial toxicity at the time of treatment [27,28]. In addition, based on reports that health insurance literacy is associated with financial hardship [29], it may be important to improve patient literacy as a solution to financial toxicity. It has been suggested that patient–physician communication regarding treatment costs may reduce the financial burden [30], but there is limited mention of patient–physician discussion of the financial burden in guidelines [31]. Further research is necessary to investigate the practicality or relevance of these matters, and to elucidate the global impact of disease in terms of financial effort, not only for patients but also for medical institutions [32,33].

This study had some limitations. In total, 508 patients were sent e-mails requesting survey responses irrespective of treatment history, and there were 101 responses from patients who received systemic treatment, resulting in a 19.9% response rate. The mean age of the patients in this study was 44.2 years, which is much younger than the mean age of mRCC patients in Japan of 66 years [20]. The low response rate to the questionnaire and the age differences observed in this study may be attributed to the fact that the research was conducted online. A study of response rates to the Japanese census showed that while approximately 60% of respondents up to the age of 40 responded via the Internet, only approximately 10–40% of those aged 60 and older did so [34]. In contrast, the percentage of respondents who responded by filling out a paper survey form was approximately 30% for those in their 40s and approximately 50–60% for those in their 60s and older. These results indicate that the response rates for online surveys tend to be higher among the younger age groups, whereas the response rates for paper-based surveys increase with age. Other studies have also reported similar trends [35]. It is recommended that future surveys gather data from a broader range of patients, including older individuals, to avoid possible age bias in the sampling of mRCC patients in Japan. Because this was a web-based survey study, the number of patients who received systemic treatment in survey panels was limited, and the proportion of patients who did not report “household income/savings” was high; thus, it is desirable to increase the number of mRCC patients in future surveys. 

## 5. Conclusions

This study revealed that financial toxicity impacts mRCC patients and affects their QOL, even though the universal health insurance system and the high-cost medical care system in Japan limit the co-payment of drug costs to some extent. It is necessary to recognize the existence of financial toxicity in these patients and communicate appropriately with them to aim for treatment that best maintains their QOL. Further studies are needed to better understand the burdens of treatment costs in larger samples of mRCC patients.

## Figures and Tables

**Figure 1 cancers-16-01904-f001:**
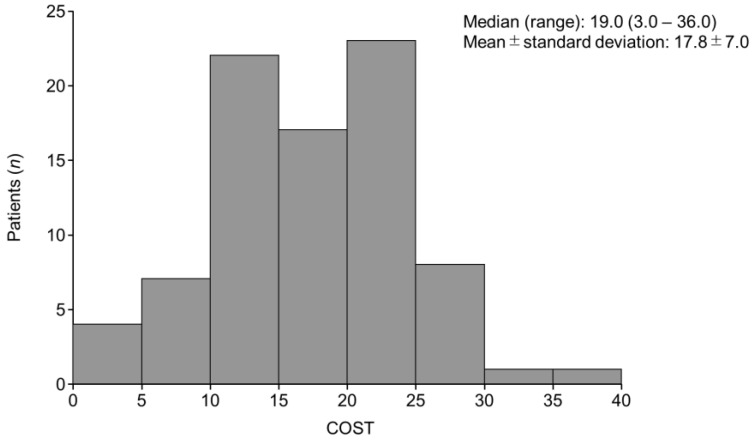
Distribution of COST scores. COST, COmprehensive Score for financial Toxicity.

**Figure 2 cancers-16-01904-f002:**
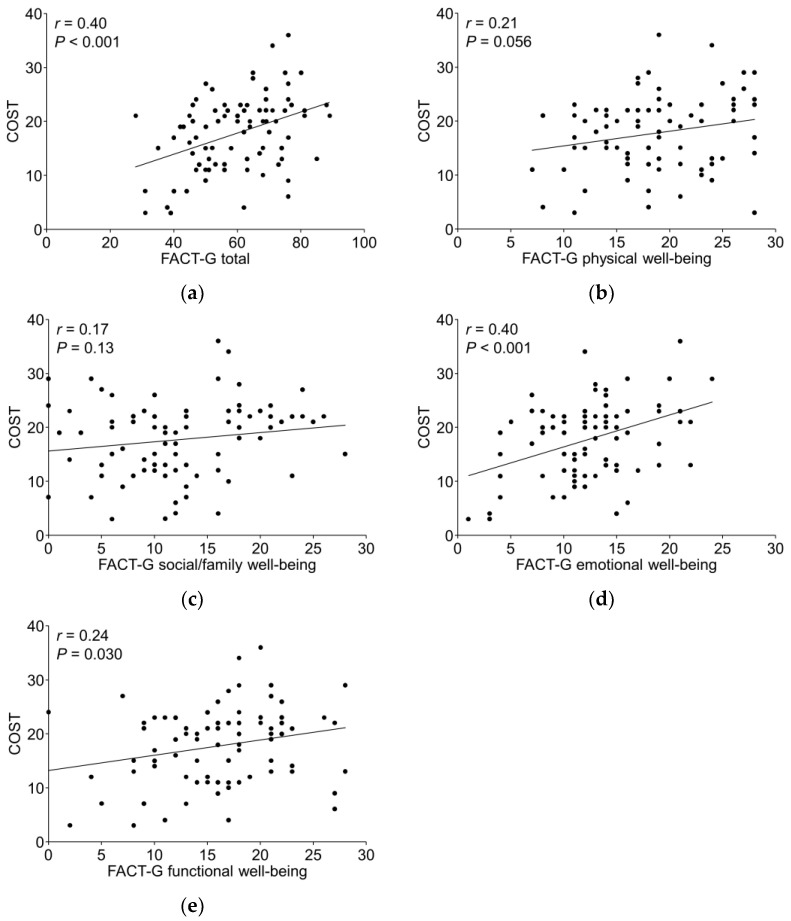
Correlation between COST and FACT-G and FACT-G sub-scales. (**a**) Correlation between COST and FACT-G. (**b**) Correlation between COST and physical well-being FACT-G subscale. (**c**) Correlation between COST and social/family well-being FACT-G subscale. (**d**) Correlation between COST and emotional well-being FACT-G subscale. (**e**) Correlation between COST and functional well-being FACT-G subscale. FACT-G, Functional Assessment of Cancer Therapy-General.

**Table 1 cancers-16-01904-t001:** Demographic characteristics of the study participants.

Characteristics	*n* = 83
Age (mean ± SD), years	44.2 ± 12.2
Gender, *n* (%)	
Man	53 (63.9)
Woman	30 (36.1)
Treatment history, *n* (%)	
Systemic therapy	83 (100)
Surgery	31 (37.3)
Radiation therapy	3 (3.6)
Duration of systemic therapy, *n* (%)	
<6 months	25 (30.1)
6 months−2 years	29 (34.9)
>2 years	29 (34.9)
Employment status, *n* (%)	
Regular employee	50 (60.2)
Self-employed	7 (8.4)
Office worker	42 (50.6)
Civil servant	1 (1.2)
Non-regular employee	33 (39.8)
Part-time	7 (8.4)
Homemaker	16 (19.3)
Unemployed	6 (7.2)
Other	4 (4.8)
Private health insurance, *n* (%)	
Yes	35 (42.2)
No	48 (57.8)
Household income per year (¥), *n* (%)	
<2,000,000	3 (3.6)
2,000,000−3,999,999	9 (10.8)
4,000,000−5,999,999	14 (16.9)
6,000,000−7,999,999	7 (8.4)
8,000,000−9,999,999	5 (6.0)
≥10,000,000	8 (9.6)
Not reported	37 (44.6)
Household savings (¥), *n* (%)	
<2,000,000	17 (20.5)
2,000,000−3,999,999	4 (4.8)
4,000,000−5,999,999	3 (3.6)
6,000,000−7,999,999	1 (1.2)
8,000,000−9,999,999	3 (3.6)
10,000,000−14,999,999	2 (2.4)
≥15,000,000	8 (9.6)
Not reported	45 (54.2)

¥, Japanese yen; SD; standard deviation.

**Table 2 cancers-16-01904-t002:** Linear regression analysis of the COST.

Variable	*n*	Univariate Analysis		Multivariate Analysis	
		*β* (95% CI)	*p* Value	*β* (95% CI)	*p* Value
Age (years)					
<65	75	−3.42 (−8.56 to 1.72)	0.19	−5.21 (−10.4 to −0.0279)	0.049
≥65	8	Ref.		Ref.	
Treatment history					
Surgery + radiation therapy + systemic therapy	3	−5.86 (−14.3 to 2.57)	0.17	−4.11 (−12.0 to 3.78)	0.30
Surgery + systemic therapy	28	Ref.		Ref.	
Systemic therapy only	52	0.220 (−3.03 to 3.47)	0.89	−0.632 (−4.06 to 2.79)	0.71
Duration of systemic therapy					
<6 months	25	2.56 (−1.13 to 6.25)	0.17	1.78 (−2.27 to 5.82)	0.38
6 months−2 years	29	−2.14 (−5.69 to 1.42)	0.24	−1.87 (−5.47 to 1.74)	0.30
>2 years	29	Ref.		Ref.	
Employment status					
Regular employee	50	Ref.		Ref.	
Non-regular employee	33	−1.25 (−4.37 to 1.87)	0.43	−2.89 (−5.93 to 0.149)	0.062
Private health insurance					
Yes	35	Ref.		Ref.	
No	48	−4.23 (−7.19 to −1.27)	0.0057	−4.90 (−8.02 to −1.79)	0.0024
Household income per year (¥)					
<2,000,000	3	−11.7 (−20.7 to −2.68)	0.012	N/A	
2,000,000−3,999,999	9	−12.3 (−18.7 to −5.78)	0.00045		
4,000,000−5,999,999	14	−9.52 (−15.4 to −3.61)	0.0023		
6,000,000−7,999,999	7	0.625 (−6.28 to 7.53)	0.86		
8,000,000−9,999,999	5	−0.975 (−8.58 to 6.63)	0.80		
≥10,000,000	8	Ref.			
Household savings (¥)					
<2,000,000	17	−5.29 (−17.5 to 6.93)	0.38	N/A	
2,000,000−3,999,999	4	0.00 (−14.2 to 14.2)	1.0		
4,000,000−5,999,999	3	−7.00 (−21.9 to 7.93)	0.35		
6,000,000−7,999,999	1	0.00 (−20.0 to 20.0)	1.0		
8,000,000−9,999,999	3	−4.33 (−19.3 to 10.6)	0.56		
10,000,000−14,999,999	2	Ref.			
≥15,000,000	8	5.75 (−7.18 to 18.7)	0.37		

## Data Availability

The data presented in this study are available on request from the corresponding author.

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
