# Peer review of "Financial Toxicity in Japanese Patients with Metastatic Renal Cell Carcinoma: A Cross-Sectional Study"

_cancers, 2024, doi:10.3390/cancers16101904_

Round 1

Reviewer 1 Report (Previous Reviewer 1)

Comments and Suggestions for Authors

Go Kimura  and co-authors present a high quality and well-written cross-sectional study focused on financial toxicity in Japanese patients with metastatic renal cell carcinoma.

Authors aimed to evaluate financial toxicity for Japanese mRCC patients using the COST tool through a web-based survey, and assess QOL using the FACT-G.

Authors conducted a web-based survey to evaluate the financial toxicity experienced by Japanese mRCC patients using the COmprehensive Score for financial Toxicity (COST) tool. This study enrolled Japanese patients who underwent, or were undergoing, systemic therapy for mRCC. The outcomes evaluated were the distribution of COST scores, the correlation between COST and quality of life (QOL) assessed by the Functional Assessment of Cancer Therapy - General (FACT-G) scale, and demographic factors associated with financial toxicity. The median (range) COST score was 19.0 (3.0–36.0). The Pearson correlation coefficient for COST and FACT-G total scores was 0.40. Univariate analysis revealed that not having private health insurance and lower household income per year were significantly associated with lower COST scores. Multivariate analyses showed that age < 65 years and not having private health insurance were significantly associated with lower COST scores. This study revealed that Japanese mRCC patients experience adverse financial impacts even under the universal health insurance coverage system available in Japan, and financial toxicity negatively affects their QOL.

Authors claim that this study is the first to identify factors affecting financial toxicity for Japanese patients with drug-treated mRCC using the COST tool. In a previous study of patients diagnosed with mRCC in Japan between 2008 and 2018, the mean age of patients was reported to be 66 years (range: 37–87 years). In comparison, patients in the present study were recruited via the Internet and thus included a younger patient population, with a mean age of 44.2 years. In this age group, illness is considered to have a greater impact than in other age groups because of how it affects work and child-rearing, so this study is unique in that it analyzed financial toxicity in this age group. 

Finally, authors conclude that they revealed that financial toxicity impacts mRCC patients and affects their QOL even though the universal health insurance system and the high-cost medical care system in Japan limit the co-payment of drug costs to some extent.

Overall, the manuscript is highly valuable for the scientific community and should be accepted for publication.

======================

Other comments to authors:

1) Please check for typos throughout the manuscript.

2) Please improve figures/tables where appropriate.

Author Response

We thank the Reviewer for their positive appraisal of our manuscript.

We have checked the manuscript throughout and made any necessary improvements.

Reviewer 2 Report (New Reviewer)

Comments and Suggestions for Authors

I would like to congratulate the Authors for their excellent and original work on a specific theme that represents a topic of relevance for both the pharmacology, financial and urology communities. Results and interpretation from the analysis of financial toxicity in Japanese patients with metastatic renal cell carcinoma represent indeed an always important argument to further assess. Nevertheless, the article in its current form deserves further revision before being considered suitable for publication. 

First and more importantly, the introduction is too extensive, going into details that can distract lectors from the main focus which sholud be the association between financial efforts and medial costs of mRCC regardless age and employment of patients. What is more there is no mention about any financial support by the country which can change the point of view. The methods of the article are too extensive, but good organized in the attempt to have an overall clarity of the manuscript. The results are clear and well organized with a good comprehension by readers. Additionally, even if the article led to significant results and apply significance to the field of interest the background and the discussion of the document remain far from the readership which may be not familiar with the topic with too generalization. For this reason, I would suggest focusing the argument regarding the association between general medical costs and helth care, adding information about worldwide actual situation, financial support by any each country and comprison with other country welfare. What is more can be done a deeper analysis on the global impact of disease in terms of financial effort not only for patients but also for the centers, like in (DOI: 10.1016/j.euf.2024.02.004) were is analized the general financial effort.

I please kindly ask the reviewer to address their minor revision and provide a document with the track changes of the methods, discussion, and references in order to facilitate the review process. 

I look forward to seeing the updated version of this manuscript

Comments on the Quality of English Language

I do not find any difficult in the comprehension of text,  reason why I confirm a good english quality

Author Response

First and more importantly, the introduction is too extensive, going into details that can distract lectors from the main focus which sholud be the association between financial efforts and medical costs of mRCC regardless age and employment of patients.

Response: Thank you for your attention to our manuscript. The purpose of this study was to evaluate the financial toxicity of mRCC patients in Japan using the COST tool, rather than to identify the medical costs paid by mRCC patients themselves or the relationship between financial effort and medical costs. Financial toxicity does not just refer to medical costs, but is a term that includes the negative impact on the patient and their family due to the financial burden associated with being diagnosed with cancer. The COST tool is used to assess this (cited in ref. 10), and so to help readers who are not familiar with it, the Introduction describes the COST tool and the relevant information surrounding it. Therefore, we feel that the current content of the Introduction is appropriate and no changes have been made.

What is more there is no mention about any financial support by the country which can change the point of view.

Response: This report focuses on Japan as the country of interest. As such, we have described the financial support context for healthcare in Japan on lines 78–83 of the Introduction.

The methods of the article are too extensive, but good organized in the attempt to have an overall clarity of the manuscript. The results are clear and well organized with a good comprehension by readers.

Additionally, even if the article led to significant results and apply significance to the field of interest the background and the discussion of the document remain far from the readership which may be not familiar with the topic with too generalization. For this reason, I would suggest focusing the argument regarding the association between general medical costs and health care, adding information about worldwide actual situation, financial support by any each country and comparison with other country welfare.

Response: Thank you for your comments on this matter. The purpose of this study was not to investigate general medical costs, but to specifically investigate financial toxicity using the COST tool. Therefore, we focused on COST scores rather than general medical costs in the Discussion. A comparison of COST scores between Japan and other countries is provided in the Discussion (lines 255–265). We have also made a comparison with the U.S. (ref. 21), where public health insurance system coverage is limited to elderly, disabled, and low-income people, and most people who do not meet these conditions use private health insurance. The most important point of this paper is that even though co-payments are kept low in Japan, compared with other countries, financial toxicity still exists.

What is more can be done a deeper analysis on the global impact of disease in terms of financial effort not only for patients but also for the centers, like in (DOI: 10.1016/j.euf.2024.02.004) were is analyzed the general financial effort.

Response: Thank you very much for referring us to this interesting paper, which is excellent because it reveals the impact of specific conditions and treatments on healthcare costs. Unfortunately, however, we are unable to perform the same analysis in this study because we did not examine the actual medical costs for each condition or treatment as in that paper. The importance of such an analysis has been added to the Discussion as an issue for future research, and two references have been added.

Reviewer 3 Report (New Reviewer)

Comments and Suggestions for Authors

The cost of advanced cancer treatment has been notably increasing with the advent of modern drugs, including molecular targeted drugs and immunooncology drugs. This trend is also observed in metastatic renal cell carcinoma (mRCC). Financial toxicity, a recently introduced term, describes the personal financial burden faced by patients receiving cancer treatment. However, financial toxicity for mRCC has not been thoroughly investigated, particularly in Japan, which has a unique medical insurance system.

In this study, the authors aimed to evaluate the financial burden of treatment for mRCC and identify factors that influence it. The Comprehensive Score for Financial Toxicity (COST) was used to assess financial toxicity, while the Functional Assessment of Cancer Therapy - General (FACT-G) was used to evaluate the quality of life (QOL) of the patients.

The authors conducted an Internet survey, sending an email to 508 mRCC patients. They received replies from 101 patients who had received systemic treatment, resulting in a survey response rate of 19.9%. Finally, 83 patients were analyzed in the study. Through multivariate analysis the authors found that age < 65 years and not having private health insurance were significantly associated with lower COST scores, indicating higher financial toxicity.

The manuscript includes two figures, along with an additional supplementary figure, and it contains two comprehensive tables.

It's worth noting that this study utilized an Internet survey, and the mean age of the patients analyzed was 44.2 years, which is younger than the mean age of general mRCC patients in Japan. This suggests that this cohort may not be fully representative of the general mRCC patient population.

Nevertheless, this study is thorough and meticulous, addressing an underreported aspect of cancer treatment. The manuscript can be accepted in its present form.

Author Response

We thank the Reviewer for their positive appraisal of our manuscript.

Round 2

Reviewer 2 Report (New Reviewer)

Comments and Suggestions for Authors

Thank you for your answer. I understand the focus of your analysis and I think that is suitable for the readers. The paper is now clear and totally focused. I appreciate the addition of the references suggested which have been considered interesting for you and improve the focus of your paper.

This manuscript is a resubmission of an earlier submission. The following is a list of the peer review reports and author responses from that submission.

Round 1

Reviewer 1 Report

Comments and Suggestions for Authors

Go Kimura and co-authors present a high quality and well-written cross-sectional study focused on financial toxicity in Japanese patients with metastatic renal cell carcinoma.

Authors aimed to evaluate financial toxicity for Japanese mRCC patients using the COST tool through a web-based survey, and assess QOL using the FACT-G. 

Authors evaluated the financial toxicity experienced by Japanese mRCC patients using the COmprehensive Score for financial Toxicity tool. This study enrolled Japanese patients who underwent, or were undergoing, systemic therapy for mRCC. The outcomes evaluated were the distribution of COST scores, the correlation between COST and quality of life assessed by the Functional Assessment of Cancer Therapy - General scale, and demographic factors associated with financial toxicity. The median (range) COST score was 19.0 (3.0–36.0). The Pearson correlation coefficient for COST and FACT-G total scores was 0.40. Univariate analysis revealed that not having private health insurance and lower household income per year were significantly associated with lower COST scores. Multivariate analyses showed that age < 65 years and not having private health insurance were significantly associated with lower COST scores. 

Authors provide results on:

- Participants characteristics

- Correlation between FACT-G QOL and COST financial toxicity

- Linear regression analysis of COST

Authors revealed that Japanese mRCC patients experienced adverse financial impacts even under the universal health insurance coverage system available in Japan, and financial toxicity negatively affects their QOL.

Finally, authors conclude that financial toxicity impacts mRCC patients and affects their QOL even though the universal health insurance system and the high-cost medical care system in Japan limit the co-payment of drug costs to some extent. It is necessary to recognize the existence of financial toxicity in these patients and communicate appropriately with them to aim for treatment that best maintains their QOL. Further studies are needed to better understand the burdens of treatment cost in larger samples of mRCC patients.

Overall, the manuscript is highly valuable for the scientific community and should be accepted for publication.

======================

Other comments to authors:

1) Please check for typos throughout the manuscript.

2) Please improve figures/tables where appropriate.

Reviewer 2 Report

Comments and Suggestions for Authors

This study uses a web-based survey to explore the association between quality of life (as measured by the Functional Assessment of Cancer Therapy - General, FACT-G) and financial burden (as measured by the COmprehensive Score for financial Toxicity, COST) faced by patients who underwent or are undergoing systematic therapy for metastatic renal cell carcinoma (mRCC) in Japan. The following points will help strengthen the manuscript.

1. Line 98: Could authors first indicate the guideline that was used to report this survey research study?
2. Lines 99-115: Thanks for providing insights into the health care system in Japan in the introduction section. Please consider a concise setting section here, to explain how care for those with mRCC is organized in Japan. How many of those “21,000 patients with RCC” (line 56) in Japan underwent systematic therapy for mRCC in “2019”? What were the breakdowns for different types of treatment modality in 2019? How many patients had private insurance? What were the aggregate expenditures in 2019 for these patients? How were the expenditures distributed across “national health insurance”, “out-of-pocket”, “High-cost Medical Expense Benefit System” for these patients…
3. Lines 99-115: This reviewer is concerned that many of the necessary components of survey research methodology is missing in this section (including a sample size calculation… random sample selection procedures used in survey administration, similarly the number of attempts made to contact responders, the calculation of response rates… the evidence of the representativeness of the study sample of the population it is targeting…)
4. Lines 148-149: Please explain how a sample size can be determined by “the number of patients registered in the web-based survey panels”? What is the theoretical foundations of this “sample size”? What is the basis for the “target number” of “95 patients”?
5. Lines 161-177 & Table 1: In general, this section should start with providing information about the population of interest in Japan and report evidence about how representative the study sample is of the population for the following results to be meaningful… if the sample is not representative, reasons should be provided to explain why it is not representative and the mitigation strategies used should be discussed…. Table 1 should include population characteristics side-by-side with the study sample characteristics.
6. Lines 232-233: “ … [For] Japanese patients”? What is the evidence?
7. Lines 233-235: Again, what evidence do authors have in that these survey results are representative of a “young patient” population of mRCC in Japan?
8. Lines 295-296: Again, what are the general characteristics of mRCC population in Japan? What is the mean age of mRCC patients in Japan? Is age the only characteristic that is not representative in this study sample?

Reviewer 3 Report

Comments and Suggestions for Authors

Taking a more measured approach, it's clear that while the study titled "Financial Toxicity in Japanese Patients with Metastatic Renal Cell Carcinoma: a Cross-sectional Study" has its merits, there are significant areas that need improvement:

  • Increasing Participant Numbers: It would be highly beneficial for the study to increase its sample size. A larger group of participants would enhance the statistical power and generalizability of the findings, offering a more robust picture of the financial toxicity in this patient population.
  • Addressing Survey Bias in Limitations Section: The limitations section of the manuscript should be thoroughly updated to acknowledge and discuss the selection bias inherent in the survey methodology. This would demonstrate an awareness of the study's constraints and provide context for interpreting the results.
  • Improving Structure and Clarity: The overall structure of the paper requires significant improvement. This includes ensuring that each section of the manuscript is clearly defined and contributes effectively to the overall narrative and flow of the paper. The arguments and data should be presented in a logical and coherent manner.
  • Condensing Introduction and Discussion: The introduction and discussion sections should be more concise. Currently, these sections may contain redundant information, which could detract from the paper's overall impact. A focused and streamlined approach would make the paper more reader-friendly and emphasize the key points more effectively.

Comments on the Quality of English Language

check typos 
